# No Association between 25-Hydroxyvitamin D and Insulin Resistance or Thyroid Hormone Concentrations in a Romanian Observational Study

**DOI:** 10.3390/medicina57010025

**Published:** 2020-12-30

**Authors:** Roxana Adriana Stoica, Cristian Guja, Anca Pantea-Stoian, Raluca Ioana Ștefan-van Staden, Ioana Popa-Tudor, Simona Diana Ștefan, Robert Ancuceanu, Cristian Serafinceanu, Constantin Ionescu Tîrgoviște

**Affiliations:** 1Departament of Diabetes, Nutrition and Metabolic Diseases, University of Medicine and Pharmacy Carol Davila, 020475 Bucharest, Romania; cristian.guja@umfcd.ro (C.G.); anca.stoian@umfcd.ro (A.P.-S.); simona.stefan@drd.umfcd.ro (S.D.Ș.); cristian.serafinceanu@umfcd.ro (C.S.); cit.paulescu@gmail.com (C.I.T.); 2Laboratory of Electrochemistry and PATLAB, National Institute of Research for Electrochemistry and Condensed Matter, 060021 Bucharest, Romania; raluca_ioana.van@upb.ro (R.I.Ș.-v.S.); johana.popa@gmail.com (I.P.-T.); 3Departament of Botanical Pharmaceutics, University of Medicine and Pharmacy Carol Davila, 020956 Bucharest, Romania; robert.ancuceanu@umfcd.ro

**Keywords:** 25-hydroxyvitamin D, insulin resistance, HOMA-IR, QUICKI

## Abstract

*Background and objectives:* Vitamin D is involved in insulin resistance through genomic and non-genomic mechanisms. Several observational and randomized studies have discrepant results; some of them showed an improved insulin resistance (IR), and others a neutral effect after vitamin D deficiency is corrected. *Materials and Methods:* We designed a retrospective observational study that included all women who presented for 33 months in an outpatient clinic in Bucharest, Romania. *Results:* We analyzed 353 patients with a mean age of 58.5 ± 13.7 years, a mean body mass index (BMI) of 27.36 ± 4.87 kg/m^−2^, and a mean level of 25-hydroxyvitamin D (25OHD) of 39.53 ± 15.73 ng/mL. There were no differences in the calculated Homeostatic Model Assessment of Insulin Resistance variants 1 and 2 (HOMA-IR) and the Quantitative Insulin Sensitivity Check Index (QUICKI) between women with vitamin D deficit versus normal values. In multivariate analysis, there was no significant relation between 25OHD and the response variables considered by us. *Conclusions:* We observed a small positive correlation between a higher level of 25OHD and increased glycosylated hemolobin (HbA1c) or IR indices without clinical significance. Other modifiable or non-modifiable factors override 25OHD influence on IR in adult women with a normal serum level and may contribute to the remainder of the variability observed.

## 1. Introduction

The prevalences of prediabetes and thyroid diseases have increased substantially in the last decade [1]. Specialists in endocrinology frequently encounter them in association with obesity, dyslipidemia and vitamin D deficiency [2]. All these factors may contribute to insulin resistance (IR) and compromise beta-cell function, leading eventually to diabetes mellitus [3].

A series of observational studies identified either a positive or a neutral effect [4,5,6,7,8,9] of vitamin D administration on lipid profile (decreasing low-density lipoprotein cholesterol-LDLc, raising high-density lipoprotein cholesterol-HDLc) or insulin resistance markers like the Homeostatic Model Assessment of Insulin Resistance (HOMA-IR) and the Quantitative Insulin Sensitivity Check Index (QUICKI). Likewise, randomized clinical studies have contrary results [10,11,12,13,14,15,16]. The vitamin D effect is challenging to isolate, being dependent on the environment, the patient’s lifestyle, comorbidities, supplementation dose, and duration of the studies [17].

A meta-analysis of observational studies identified a type 2 diabetes (T2DM) risk reduction of 4% for each 10 nmol/L increase in 25-hydroxyvitamin D (25OHD) level [18]. This trend is not visible in all studies. In patients with prediabetes and suboptimal levels of 25OHD, the supplementation for 26 weeks in a randomized clinical trial did not improve peripheral or hepatic IR evaluated by a euglycemic-hyperinsulinemic clamp [19]. Also, co-supplementation with vitamin D and omega-3 fatty acid in reproductive-age women improved fasting blood glucose and HOMA of beta-cell function, but without significant differences between groups [20].

A post hoc analysis of three extensive observational studies did not demonstrate an increased incidence in T2DM, regardless of the low vitamin D levels in women with a mean age of 66 years [21]. In another analysis of subpopulations in the Hoorn study, there was no significant association between 25OHD and T2DM [22].

All these clinical studies are based on the link between vitamin D and IR explained by genomic and non-genomic mechanisms [23]. The active form of this hormone-1,25-hydroxyvitamin D (1,25OHD) has receptors on the pancreatic beta-cell, and on insulin-dependent tissues (adipose, muscular and hepatic). At the beta cell level, the coupling of 1,25OHD to its specific receptors (Vitamin D Receptor—VDR) initiates the pathogenic cascade, followed by the intracellular binding with retinoid receptors. The complex is translocated at the nuclear level, where it binds to the responsive elements (VDRE) [24]. Stimulation of the VDRE induces the transcription of the insulin gene and other factors that are involved in cytoskeleton organization [25]. Also, it determines the intracellular calcium elevation and insulin vesicles exocytosis [24].

In the peripheral insulin-dependent tissues, calcium is the common mediator for the two hormones. When vitamin D deficiency is established, the level of intracellular calcium is higher, similar to the beta cell, although the serum value may not be changed. This translates into a decreased activity of glucose transporter 4 (GLUT-4) and IR [26].

Scarce data exist for the relationship between 25OHD, IR, and frequently associated conditions like thyroid disease in Romanian Caucasian adult populations. The novelty of our study resides in the analysis of vitamin D status in a real-life adult female population with multiple endocrinological pathologies. Although previous studies tend to address them separately, we preferred a cross-sectional design that allowed us to include multiple laboratory data to generate hypotheses about the relationship between biological parameters.

## 2. Materials and Methods

This is a retrospective observational study that included female patients seen by two endocrinologists in an outpatient clinic in Bucharest, Romania. The reasons for referral to the endocrinologist were routine visits for thyroid diseases or osteoporosis. All patients were analysed by the same laboratory for hormonal and biochemical blood tests between July 2017 and March 2020. We considered the following variables: glycemia (mg/dL), glycosylated hemolobin (HbA1c) (%), total cholesterol (mg/dL), high-density and low-density lipoprotein cholesterol–HDLc, LDLc (mg/dL), tryglicerides (TG) (mg/dL), thyroid stimulating hormone or thyrotropin–TSH (μUI/mL), thyroxine-fT4 (ng/dL), parathormone- PTH (ng/dL), total calcium (mg/dL), magnesium (Mg) (mg/dL), phosphor (mg/dL), 25-hydroxyvitamin D (25OHD) (ng/mL), fasting insulin (μUI/mL) and fasting C-peptide (ng/dL). The optimum level of 25OHD was considered 30–100 ng/mL, vitamin D insufficiency between 21 and 29 ng/mL, and deficiency below 20 ng/mL [27].

We collected the blood test results and data regarding age, weight, height, and comorbidities. We formed our database based on laboratory data, and details about the treatment and supplementation were not available. The body mass index (BMI) was calculated as weight (kg)/height(m)^−2^. For estimating IR we used: HOMA-IR1 (insulin) = fasting glycemia (mg/dL) * fasting insulin (μUI/mL)/405, HOMA-IR1 (C-peptide) = 1.5 + fasting glycemia (mmol/L)) × (fasting C-peptide (nmol/L)/2800 [28], and HOMA-IR2 (insulin), HOMA-IR2 (C-peptide) calculated with HOMA-IR software version 2.2.3 available online. The cut-off was set at 2.5 [29]. The QUICKI index was computed by the formula for insulin sensitivity: 1/(log(fasting insulin μU/mL) + log(fasting glucose mg/dL)) at a cut-off below 0.3 [30]. The study respected the conduct of the Helsinki Declaration. We obtained the approval of the Ethics Commission of the Diabetes Department of the University of Medicine and Pharmacy Carol Davila (National Institute of Diabetes, Nutrition and Metabolic Diseases N. C. Paulescu), located in Bucharest, Romania.

The data were introduced in a Microsoft Office Excel^®^ file (Microsoft Corp, Redmond, WA, United States, 2007) and processed with SPSS^®^ (International Business Machines Corp, North Castle, NY, United States, 2011, IBM SPSS Statistics for Windows, Version 20.0) for univariate analysis. Multivariate statistical analysis was performed in the R computing and programming environment, using the base package and additional R packages as mentioned below. Exploratory data analysis was performed using the “dlookr” [31] and “DataExplorer” [32] packages. We used multiple linear regression models to assess the association between glycemia, HbA1c, two versions of HOMA-IR, and QUICKI (as dependent variables) and plasma 25OHD (as an independent variable), adjusting for several potential confounding factors (selected based on the field knowledge as expressed in the published literature—BMI, LDLc and HDLc, TG, FT4 and TSH, serum magnesium level, and age). The assumptions of linear regression (residual normality, homoscedasticity, multicollinearity) are checked using a variety of functions available in the R packages “olsrr” [33], “ggfortify” [34], and “skedastic” [35]. When the assumptions of homoscedasticity and normality of residuals were not satisfied (due to a number of outliers), we used robust regression-based on highly efficient MM-type estimators, as implemented in the “robustbase” R package (lmrob function) [36]. Standardized coefficients were computed using the “lm.beta” [37] R package for the ordinary least squares (OLS) models and using the response and regressors standardized with the Make.z function of the “QuantPsyc” R package [38] for the robust regression models. For sensitivity analysis purposes, we considered robust regression (when the primary model was OLS), and quantile regression models for median and several quantile levels (10%, 25%, 75%, 90%), using a bootstrap procedure for the computation of standard errors (R package “quantreg” [39]). The *p*-value for statistical significance was set at 0.05.

## 3. Results

During the 33 months, 565 female patients presented for a routine visit. After excluding patients with diabetes and those with incomplete data, 353 patients were included in the final analysis. Mean age was 58.5 ± 13.7 years, with a mean body mass index (BMI) of 27.36 ± 4.87 kg/m^−2^. Population characteristics are presented in Table 1 based on vitamin D status. The mean 25OHD value of the entire population was 39.53 ± 15.73 ng/mL. Impaired fasting glucose was present in 46.45%, and 16.43% had an HbA1c between 5.7 and 6.5%. HDLc was below 50 mg/dL in 13.88% and TG above 150 mg/dL in 16.71%. Data about lipid-lowering treatment were not available.

The majority of our population presented for routine evaluation of thyroid disease. Thirty percent of the patients had autoimmune thyroiditis, 21.81% multinodular goiter, 1.98% Basedow disease, 6.52% postoperative myxedema, and the remaining percent had other pathologies like single thyroid nodule or partial agenesia. Patients were previously evaluated by echography. Osteoporosis was diagnosed in 23.51% and osteopenia in 12.18%, using osteodensitometry.

In univariate analysis, there were no significant differences regarding the IR indices between the two groups (student *t*-test for normally distributed data, respectively Mann–Whitney U-test). Correlation analysis showed a small positive relation between 25OHD and HbA1c, fasting C-peptide, HOMA-IR2 (C-peptide). A negative correlation was observed with TSH, total cholesterol and LDLc (Table 2).

Scatterplots of 25OHD and several additional variables known from the literature to influence glycemia are shown in Figure 1. The visual examination of the data suggested that there may be a weak association between 25OHD and fasting glycemia. Also, a relatively weak association was observed for HbA1c and the two different versions of HOMA-IR and QUICKI.

For fasting glycemia, there was a significant departure from normality and homoscedasticity. Therefore we used robust linear regression to describe the relationship between the independent variables of interest (BMI, LDLc, HDLc, TG, fT4 and TSH, Mg, and age) and their values. The level of 25OHD was not significantly associated with the blood glucose level (*p* = 0.08). BMI (*p* = 0.003) and age (*p* < 0.001) were significantly associated with glycemia, whereas the other variables had no significant association (Table 3). The model explains about 10% of the variability seen in glycemia (adjusted R^2^ = 10.55), whereas the rest is attributed to other variables, not included in our analysis.

By using the same set of variables to describe the variation in HbA1c, the 25OHD was significantly associated with this biomarker. After adjusting for the other covariates, the association remained positive and relatively small (*p* = 0.02, Table 4). Similarly, HDLc was positively correlated with HbA1c (*p* < 0.001). The largest effect on the HbA1c-25OHD relation was seen for BMI (*p* < 0.001), followed by HDLc (*p* <0.001), and age (*p* < 0.001). The robust regression values were very similar with respect to both standardized coefficients and *p* values, while the quantile regression values were also consistent with these results.

For HOMA-IR2 (insulin), we applied the same model, considering the number of outliers and observations with high leverage, as shown in Table 5. The 25OHD level was not significantly associated with HOMA-IR2 (insulin) (*p* = 0.24). The effect size measured by the standardized coefficient was small, and the association was positive. Only BMI (*p* < 0.001) and LDLc (*p* = 0.016) were significantly associated with this IR marker. The importance of BMI is much higher than that of LDLc (beta of 0.266, and −0.093, respectively). The quantile regression used for sensitivity analysis purposes showed that 25OHD has no significant association with HOMA-IR2 (insulin) for high quantiles (0.75, 0.95), but at lower quantiles (0.1, 0.25 and 0.5) was significantly and positively associated with it. In the quantile regression, the negative association of LDLc was significant only for the 0.1 and 0.25 quantiles. At the median and high quantiles (0.75, 0.90), the age was significantly (and positively) associated with the HOMA-IR2 (insulin) values.

Also, the regression analysis revealed that BMI was significantly associated with HOMAR-IR2 (insulin) at all quantiles evaluated. In contrast, LDLc tended to be associated with a decrease in HOMA-IR2 (insulin) only at the middle and higher quantiles.

HOMA-IR2 (C-peptide) was positively associated with the level of 25OHD (*p* < 0.001). The largest effect was observed for BMI (*p* < 0.001) and TG (*p* = 0.005), together with a weak negative association for HDLc (*p* = 0.014). In regression analysis, the only consistent associations for all quantiles were for BMI and 25OHD; the effect of HDLc was very weak (the lowest *p*-value was 0.055, recorded for the 0.25 quantile). The association between TG is consistent at higher quantiles (0.5 and higher, *p* <0.002), but not at the lower ones (0.1 or 0.25).

The 25OHD level was negatively associated with QUICKI (*p* = 0.03), similar to BMI (*p* < 0.001) and TG (*p* = 0.01) (Table 6). Apart from the small positive correlation of LDLc, none of the other covariates was significant. The quintile regression indicates that the strongest regressor is BMI (*p* < 0.001 for all quintiles), and 25OHD was significantly associated with QUICKI only for the lower quintiles, but not for those higher than the median (0.75 and 0.90). The negative effect of TG was observed across all quintiles analyzed, but it is significant only for the 0.1 and 0.75 quintiles.

## 4. Discussion

In our study, the female population had an optimal level of 25OHD (39.53 ± 15.73 ng/mL) that is different from other Romanian cohorts where a moderate deficit is present-mean level of 18.6 ng/mL [40]. This is explained by the fact that patients are closely monitored by the endocrinologist every three to six months, and some of them are supplemented with vitamin D, oral calcium or magnesium. Since our database was formed by accessing laboratory data, a population bias could be considered: those investigated have a better treatment adherence and, thus, normal vitamin status. Hence, our results involving IR can be applied to females with normal vitamin D status. Treatment, food intake, solar exposure data were not available. Also, there is a seasonal variation of vitamin D [40] and IR [41] that can influence our results, but the patient’s inclusion was continuous. Also, our population was stationary and constant, the patients were included consecutively, and no reverse causality may be considered. Thus, we were able to investigate an etiological hypothesis and analyze the differences between groups in this cross-sectional design.

The relationship between 25OHD and IR is complex with many confounding factors. In our analysis, one of them was BMI that had a negative correlation with 25OHD, as previously shown [42]. In univariate analysis, we also obtained an inverse relation (negative Spearman coefficient = −0.182, *p* = 0.001) explained by the lipophilic character of 25OHD–when weight is lost, the 25OHD increases [43]. Another factor was the age that was positively correlated with 25OHD Spearman coefficient of 0.189 (*p* < 0.001). Elderly patients visit the endocrinologist more often because of associated osteopenia or osteoporosis and supplement their diet with vitamin D.

Only C-peptide and IR indices derived from it correlated with 25OHD. The explanation could be by the fact the C-peptide has a longer plasmatic half-life and it doesn’t go through hepatic extraction, thus showing better the pre-hepatic beta-cell insulin secretion, as highlighted for T2DM patients [44].

Thus, we developed a model adjusting for BMI, age, and also for lipid profile (LDLc, HDLc, TG), Mg, and thyroid function (TSH, fT4). Body composition, especially body fat mass, and lipid profile are shown to predict prediabetes [45], so are the first to consider. Furthermore, we chose magnesium over calcium because low levels are associated with an increased risk for prediabetes and diabetes, mechanisms partly involving IR [46]. Vitamin D acts on raising the intracellular calcium levels and stimulating insulin secretion from the beta cells. However, variations of plasmatic calcium are very narrow and attenuated by the influence of other hormones [47]. The previous meta-analysis suggest that a low 25OHD is associated with thyroid disease, in particular autoimmune thyroiditis [48]. After adjusting for all these factors, there is a small positive correlation between HbA1c and 25OHD, as opposed to the results of Nur-Eke et al. [49].

In light of other observational data that claimed a negative relationship between HbA1c and 25OHD [50], this is a surprising result but is in agreement with other randomized clinical trials that found no effect of the vitamin D3 supplementation on HbA1c [12,51]. Even in observational studies with benefits claimed for 25OHD, the correlation between 25OHD and HbA1c was relatively small. For instance, a correlation coefficient of −0.16 is reported by Buhary et al. [50], corresponds to a coefficient of determination of 0.0256, which means that 25OHD would explain about 2.56% of the variability seen in HbA1c. Moreover, this study only used a univariate analysis and, therefore, confounding variables might have contributed to the correlation observed.

We used the same confounding factors in the IR analysis. The most positive influence is from BMI, followed by HDLc, and age. There is no additional advantage of a normal 25OHD level for the reduction of IR. Other factors override the benefit of vitamin D supplementation on glycemic values, like BMI, age, and some not studied by us like diet, inflammation, and sex hormones. It is well established that a decrease in BMI and abdominal fat determine IR reversal as shown by O’Leary et al. [52] in a population with the same mean age as ours. Advancing age is associated with sarcopenia and abdominal adiposity that contributes to metabolic disturbances, that might be more important than 25OHD deficit per se.

However, there are no significant differences between patients with 25OHD deficit versus normal, total cholesterol and LDLc are higher in the first group. These two variables correlate inversely with 25OHD. This opens the discussion on the extended cardiovascular roles of 25OHD. Observational studies described that vitamin D deficiency is associated with an increased cardiovascular risk, an increased mortality, irrespective of parathyroid hormone (PTH) [53,54]. The mechanism could involve a pro-atherogenic lipid profile modification even in healthy middle aged women as reported by Giovinazzo et al. [55]. In their study, HDLc was positively correlated with vitamin D [55], as opposed to our results where LDLc seem to be more important.

In the study of Yang Y et al. [56], there was no independent effect of 25OHD on HOMA-IR or HOMA-B. Nonetheless, a significant correlation with the early-phase insulin secretion index and area under the insulin curve emerged. These indices were unavailable in our analysis because we assessed only the fasting blood tests. Moreover, in a euglycemic hyperinsulinemic clamp study by Wallace et al. [19], with similar population characteristics to our study, except the presence of thyroid disease, there was no association between 25OHD and IR.

When other risk factors are not modified, a sufficient 25OHD level is not enough to reduce IR and change the type 2 diabetes debut. This can explain why elevated 25OHD values are positively associated with high HOMA-IR2 (C-peptide) and low QUICKI index. Results from other observational studies of Romanian Caucasian postmenopausal women have consistent results [41]. In the Women’s Health Initiative study, the supplementation with vitamin D and calcium did not reduce diabetes risk [57], and 25OHD alone was not associated with an increased risk for this disease [21]. Contrariwise, a systematic review with several observational studies concluded that a low 25OHD elevates diabetes risk [23]. Our results are congruent with the randomized clinical study that demonstrated a neutral effect of supplementation, especially when it is done at a normal serum value [12].

The strengths of our study are the analysis of 25OHD level in relation to thyroid function and phosphocalcic metabolism. Our patients have associated thyroid disease, most of them with autoimmune thyroidis and multinodular goiter. The single positive association was between TSH and HbA1c without clinical significance. Total calcium, magnesium, phosphorus and PTH do not influence IR in our sample (data not shown). Another strength is the uniformity of the blood tests, being analyzed by the same laboratory. One of the limits of this study is that we could not control the exposure, meaning the vitamin D supplementation, diet, physical activity. Also, we analyze only the female population because the proportion of men with 25OHD determination was very small and was excluded. Our study is cross-sectional and, therefore, an improvement in HbA1c after the baseline deficit of 25OHD is corrected may not be excluded. Furthermore, there are other IR or the early phase of insulin secretion indices that were not explored by us. There was considerable variability among the subjects, and the variables included in our model only explained at most 30% of the variability recorded.

## 5. Conclusions

There were no significant associations between 25OHD and the response variables considered by us. We observed a small positive correlation between a higher level of 25OHD and an increase in HbA1c, HOMA-IR2 (C-peptide), as well as a decrease in QUICKI. BMI had the strongest association with the response variables considered. Other modifiable or non-modifiable factors override 25OHD influence on IR in adult women with a normal serum level and may contribute to the remainder of variability observed in our study. Supplementation is recommended in patients with vitamin D deficit, and further studies are needed to demonstrate the pleiotropic effects in real-life cohorts that frequently have associated endocrine diseases.

## Figures and Tables

**Figure 1 medicina-57-00025-f001:**
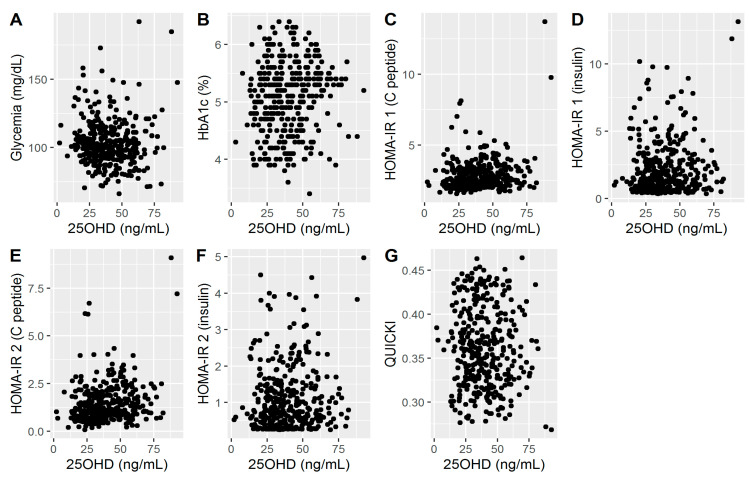
(**A**). Associations between 25OHD and glycemia; (**B**). Associations between 25OHD and HbA1c; (**C**). Associations between 25OHD and HOMA-IR1 (C peptide); (**D**). Associations between 25OHD and HOMA-IR1 (insulin); (**E**). Associations between 25OHD and HOMA-IR2 (C peptide); (**F**). Associations between 25OHD and HOMA-IR2 (insulin); (**G**). Associations between 25OHD and QUICKI.

**Table 1 medicina-57-00025-t001:** Population characteristics based on 25-hydroxyvitamin D (25OHD) levels.

Blood Test	25OHD < 30 ng/mL (*n* = 109)	25OHD > 30 ng/mL (*n* = 244)
Age (years)	61 ± 10.5	62 ± 19
BMI (kg/m^−2^)	28.33 ± 5.03	26.92 ± 4.75
Glycemia (mg/dL)	104.78 ± 14.97	102.71 ± 17.64
HbA1c (%)	5.1 ± 0.9	5.2 ± 0.9
Total cholesterol (mg/dL)	211.18 ± 43.89	196.03 ± 39.63
HDLc (mg/dL)	66.00 ± 29.5	65.30 ± 24.0
LDLc (mg/dL)	113.94 ± 38.44	105.84 ± 35.72
TG (mg/dL)	97.61 ± 63.24	97.56 ± 55.72
TSH (μUI/mL)	1.82 ± 2.02	1.68 ± 1.51
fT4 (ng/dL)	1.15 ± 0.24	1.13 ± 0.21
PTH (ng/dL)	40.33 ± 20.18	37.50 ± 19.50
Total calcium (mg/dL)	9.49 ± 0.59	9.65 ± 0.55
Magnesium (mg/dL)	2.04 ± 0.20	2.02 ± 0.31
Phosphorus (mg/dL)	3.76 ± 0.96	3.72 ± 0.97
IR indices		
Insulin (μUI/mL)	5.59 ± 6.62	6.68 ± 7.27
C-peptide (ng/dL)	1.42 ± 1.21	1.83 ± 1.37
HOMA-IR1 (insulin)	1.42 ± 1.93	1.61 ± 2.01
HOMA-IR2 (insulin)	0.74 ± 0.89	0.88 ± 0.99
HOMA-IR1 (C-peptide)	2.47 ± 0.9	2.63 ± 1.12
HOMA-IR2 (C-peptide)	1.08 ± 0.86	1.33 ± 1.1
QUICKI	0.36 ± 0.07	0.36 ± 0.06
Triglyceride/HDLc ratio	1.46 ± 1.68	1.44 ± 1.11

**Table 2 medicina-57-00025-t002:** Correlation analysis for the study sample (*n* = 353).

Correlation of 25OHD with	Pearson/Spearman Coefficients	*p*-Value
Age (years)	0.189	<0.001 *
BMI (kg/m^−2^)	−0.182	0.001 *
Hba1c (%)	0.121	0.024 *
Insulin (μUI/mL)	0.057	0.287
C-peptide (ng/dL)	0.185	<0.001 *
HOMA-IR1 (insulin)	0.041	0.442
HOMA–IR2 (insulin)	0.051	0.336
HOMA-IR1 (C-peptide)	0.154	0.004 *
HOMA-IR2 (C-peptide)	0.174	0.001 *
QUICKI	−0.041	0.442
Total cholesterol (TC) (mg/dL) LDLc (mg/dL)	−0.240 −0.180	<0.001 * 0.001 *
TSH (μUI/mL)	−0.127	0.017 *
Calcium (mg/dL) Magnesium (mg/dL)	0.138 −0.05	0.01 * 0.355

* statistically significant (<0.05).

**Table 3 medicina-57-00025-t003:** A linear regression model describing the effects of 25OHD and other potentially relevant covariates on blood glucose in the study sample.

Term	Coeff.	Std. Coeff.	S.E.	*t*. Value	*p*-Value
(Intercept)	70.4367	−0.133	11.155	6.314	1.00 × 10^−9^
25OHD	−0.092	−0.086	0.051	−1.780	0.076
BMI	0.511	0.147	0.174	2.932	0.003 ***
LDLc	0.004	0.008	0.021	0.186	0.852
HDLc	−0.019	−0.021	0.042	−0.446	0.655
TG	−0.006	−0.023	0.013	−0.493	0.622
fT4	0.399	0.004	3.899	0.102	0.918
TSH	0.072	0.010	0.194	0.372	0.709
Mg	3.709	0.063	2.389	1.552	0.121
Age	0.235	0.191	0.058	4.048	6.59 × 10^−5^

Coeff-coefficient; Std. Coeff-standardized coefficients; Signif. codes: ‘***’ 0.001.

**Table 4 medicina-57-00025-t004:** A linear regression model describing the effects of 25OHD and several variables on HbA1c in the study sample.

Term	Coeff.	Std. coeff.	S.E.	*t*. Value	*p*-Value
(Intercept)	3.910	0	0.479	8.156	1.09 × 10^−14^ ***
25OHD	0.005	0.139	0.002	2.420	0.016 *
BMI	0.030	0.244	0.007	4.172	3.99 × 10^−5^ ***
LDLc	−0.001	−0.076	0.0009	−1.343	0.180
HDLc	0.007	0.243	0.001	4.078	5.88× 10^−5^ ***
TG	0.0008	0.072	0.0006	1.218	0.224
fT4	−0.326	−0.1006	0.178	−1.831	0.068
TSH	−0.015	−0.066	0.012	−1.216	0.224
Mg	−0.258	−0.122	0.111	−2.319	0.021 *
Age	0.009	0.204	0.002	3.645	0.0003 ***

Coeff-coefficient; Std. Coeff-standardized coefficients; Signif. codes: 0, ‘***’ 0.001, ‘*’ 0.05.

**Table 5 medicina-57-00025-t005:** A robust linear regression model describing the effects of 25OHD and several variables on HOMA-IR2 (insulin) in the study sample.

Term	Coeff	Std. coeff	S.E.	t. value	*p*-value
(Intercept)	−0.431	−0.245	0.522	−0.825	0.409
25OHD	0.002	0.042	0.002	1.177	0.239
BMI	0.048	0.266	0.008	5.666	3.48 × 10^−8^ ***
LDLc	−0.002	−0.092	0.0009	−2.424	0.015 *
HDLc	−0.001	−0.027	0.001	−0.671	0.502
TG	0.001	0.085	0.0008	1.532	0.126
fT4	−0.219	−0.046	0.145	−1.507	0.132
TSH	0.014	0.041	0.017	0.834	0.404
Mg	0.110	0.036	0.120	0.915	0.360
Age	0.001	0.024	0.002	0.724	0.469

Coeff-coefficient; Std. Coeff-standardized coefficients; Signif. codes: 0, ‘***’ 0.001, ‘*’ 0.05.

**Table 6 medicina-57-00025-t006:** A robust linear regression model describing the effects of 25OHD and several variables on QUICKI in the study sample.

Term	Coeff.	Std Coeff.	S.E.	t. Value	*p*-Value
(Intercept)	0.483	0.001	0.041	11.727	2.67 × 10^−26^ ***
25OHD	−0.0003	−0.120	0.0001	−2.191	0.029 *
BMI	−0.004	−0.436	0.0005	−7.000	1.74 × 10^−11^ ***
LDLc	0.0001	0.095	6.45 × 10^−5^	1.830	0.068
HDLc	3.37 × 10^−5^	0.014	0.0001	0.251	0.801
TG	−0.0001	−0.149	4.57 × 10^−5^	−2.531	0.011 *
fT4	0.021	0.089	0.016	1.326	0.185
TSH	−0.0004	−0.025	0.0008	−0.552	0.581
Mg	−0.005	−0.037	0.007	−0.784	0.433
Age	−0.0001	−0.056	0.0002	−0.895	0.371

Coeff-coefficient; Std. Coeff-standardized coefficients; Signif. codes: 0, ‘***’ 0.001, ‘*’ 0.05.

## Data Availability

Restrictions apply to the availability of these data. Data was obtained from CMI DR STOICA MARIANA laboratory database and are available the corresponding author with the permission of the first mentioned.

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
