# Peer review of "No Association between 25-Hydroxyvitamin D and Insulin Resistance or Thyroid Hormone Concentrations in a Romanian Observational Study"

_medicina, 2020, doi:10.3390/medicina57010025_

Round 1

Reviewer 1 Report

The authors present an essentially negative observational study of serum 25OHD concentration and markers of insulin resistance in 353 female outpatients seen in an endocrine clinic.  The reasons for referral to the endocrinologist are not stated, but very few of the cohort had significant 2OHVitD deficiency – which rather limits any conclusions that can be drawn.   However, across the range of 25OHVitD measured there are no striking correlations with markers of insulin resistance.   Given that this is an area of contention I think these data are of general interest – to prevent publication bias in what is a currently topical area.   I have very few comments.

The title is somewhat misleading ; “No Association between 25-Hydroxyvitamin D and Insulin Resistance or Thyroid Hormone concentrations in a Romanian Observational Study” being more representative

The presence or absence of thyroid disease is not addressed in the Population characteristics – only the serum TSH concentration.  As these are treated patients the study cannot address an association with thyroid disease. Only in the discussion is it mentioned that “Our patients have associated thyroid disease, most of them under treatment”

Introduction.  Line 37 Given the subsequent discussion around the lack of evidence for a causal relation between IR and 25OHD insufficiency I think “All these factors may contribute to insulin resistance” may be more accurate.

Line 57 – The use of pathogenic is confusion – I assume 25OHVitD deficiency may be  “pathogenic” for IR – not 25OHvit D per se

Table 1.  The IFCC recommend HbA1c should be measured in millimoles per mol (mmol/mol) not % - I’m not sure of this journals policy is regarding units as many of the analytes here are not reported in what I would consider ‘standard’ although I realise this can be journal specific.

Fig 1 The relation between serum 25OHVItD and TSH is not shown, which may be of interest given the negative correlation

Discussion.  Line 219 is rather concerning – if blood samples for Insulin were not collected in a manor that insulin is stable for analysis, this questions the entire dataset.  Some information regarding the pre-analytical sampling methods for insulin is required in the methods if this is thought to be a concern.

Author Response

Thank you for giving us the opportunity to submit our revised manuscript. The relationship between 25 hydroxyvitamin D (25 OHD) and insulin resistance is complex with many confounding factors. Our observational study comes to support the idea that in female patients with associated thyroid disease undergoing routine endocrinological visits there are no relevant correlations between these two parameters. Other factors like BMI, age, lipid profile fractions, override the importance of 25 OHD on glycemic values and insulin resistance.  We appreciate the time and effort that you have dedicated to proving your valuable feedback. We incorporated the insightful comments.

Reviewer 2 Report

The Authors retrospectively evaluated a large series of female patients suffering from multiple endocrinological disorders withtha aim to establish a correlation, if any, between their vitamin D levels and the metabolic and hormonal indices. They lack to found any significant correlation/association 

The topic of the study is no novel. A lot of data  concerning the extra-scheletric effects of vitamina D have been published in the last decades. However, since the results of the available studies are largely inconclusive, the present study may be of interest.

However, there are several major concerns that the Authors should address.

Firts of all, they claimed that the novelty of their study "redises in the analysis of vitamin D status in a real-life adult female population with multiple endocrine pathologies".

What endocrinological pathologies were the enrolled patients suffering from? Diabetes? insulin-resistance? autoimmune thyroid disease? hypothyroidism? nodules? There is no mention of this in the material and methods a section nor in the Tables or in the results section. 

The Authors should clarify what thyroid disease the patients have been diagnosed with and they should clearly report the medications they are taking. Also there is no mention of other medical therapies. 

The title contradicts the materials and methods and results section with respect to the interconnection with thyroid disease; consequently, parts of the results and discussion do not address the declared aim of the study.

The study polulation is not well characterized, and this recused the relevance of the statistical analysis.

Also, all patients were vitamin D sufficient, in constrast with data from general population. Maybe there were selection baias. I imagine the most of them were taking vitamina D supplements; this should be specified.

Moreover, in my opinion, patients with a severe  vitamina D deficiency should be considered separately from those with vitamin D insufficiency (vitamina D leves between 20 and 30 ng/ml) for statistical evaluation.

The levels of sufficiency should be set based on the
Endocrine Society guidelines [M.F. Holick, N.C. Binkley, H.A. Bischoff-Ferrari, C.M. Gordon,D.A. Hanley, R.P. Heaney, M.H. Murad, C.M. Weaver, Endocrine
Society: evaluation, treatment, and prevention of vitamin D
deficiency: an Endocrine Society clinical practice guideline.
J. Clin. Endocrinol. Metab. 96, 1911–1930 2011]

Despite the title, data concerning vitamin D and thryoid disorders are not reported either in the introduction or in the Discussion. So, please, change the title or modify the paper.

Cardiometabolic parameters should be reported and included in the statistical analysis. recently Giovinazzo and coworkers reported a significant correlation between cardio-metabolic parameters (pressure, cholesterol, IMT) in healthy post-menopausal women. The study should be quoted:J Endocrinol Invest  2017;40(12):1337-1343. doi: 10.1007/s40618-017-0707-x. Correlation of cardio-metabolic parameters with vitamin D status in healthy premenopausal women.  S Giovinazzo et al.

Author Response

We are grateful for the opportunity to further improve our manuscript. The relationship between 25 hydroxyvitamin D (25 OHD) and insulin resistance is complex with many confounding factors. Although the topic of the research is no novel, the strengths of our observational study are the relatively large number of subjects, the data uniformity – the blood samples were analyzed by same laboratory with the same methods, and the considerable number of variables assessing thyroid and phosphocalcic metabolism. This allowed us to do a complex multivariate analysis. We have made additional revisions to the manuscript following your pragmatic suggestions  and therefore you can identify the modifications written with red color.

Round 2

Reviewer 2 Report

The Authors fully met the Referees' suggestiond and requests and the overalla quality of the manuscript has been impreoved.